SOFTWARE

# `wavess`: An R package for simulation of adaptive within-host virus sequence evolution

**Narmada Sambaturu**[1,2☉], **Zena Lapp**[1☉], **Fernando D. K. Tria**[1], **Ethan Romero-Severson** [1], **Carmen Molina-París**[1], **Thomas Leitner** [1]*

**1** Theoretical Biology and Biophysics Group, Los Alamos National Laboratory, Los Alamos, New Mexico, United States of America, **2** Systems Science and Industrial Engineering, Binghamton University, Binghamton, New York, United States of America

☉ These authors contributed equally to this work.

* tkl@lanl.gov

**Data availability statement:** All data and code are available on GitHub. The package code and corresponding example data are available at

## Abstract

Simulating within-host virus sequence evolution allows for the investigation of factors such as the role of recombination in virus diversification and the impact of selective pressures on virus evolution. Here, we provide a new software to simulate virus within-host evolution called `wavess` (within-host adaptive virus evolution sequence simulator), a discrete-time individual-based model and a corresponding user-friendly R package. The underlying model simulates recombination, a latent infected cell reservoir, and three forms of selection: conserved sites fitness and replicative fitness in comparison to a reference sequence, and immune fitness including cross-reactivity imposed by a co-evolving immune response. In the R package, we also provide functions to generate model inputs from empirical data, as well as functions to analyze the simulation outputs. At user-defined time points, the software returns various counts related to the virus population(s) and a set of sampled virus sequences. We applied this model to investigate the selection pressures on HIV-1 *env* sequences longitudinally collected from 11 individuals. The best-fitting immune cost differed across individuals, mirroring the real-world expectation of heterogeneous immune responses among human hosts. Furthermore, the phylogenies reconstructed from these simulated sequences were similar to the phylogenies reconstructed from the real sequences for all summary statistics tested. To our knowledge, compared to other similar models, `wavess` has been more rigorously validated against real within-host virus sequences, and is the first to be implemented as an R package. The `wavess` R package can be downloaded from https://github.com/MolEvolEpid/wavess.

https://github.com/MolEvolEpid/wavess and the code and data to recreate the manuscript analysis and figures are available at https://github.com/MolEvolEpid/wavess_manuscript.

**Funding:** This work was supported by the National Institutes of Health [R01AI087520 to TL] and the Los Alamos National Laboratory [Laboratory Directed Research and Development program fellowship project no. 20230873PRD4 to ZL and project no. 20210959PRD3 to NS]. The funders had no role in study design, data collection and analysis, decision to publish, or preparation of the manuscript.

**Competing interests:** The authors have declared that no competing interests exist.

## Author summary

During a virus infection, the virus population within an individual has the potential to evolve over time. While the evolutionary rate and duration of infection determine the number of mutations the virus can accumulate, changes to a virus sequence may impact protein function and host immune recognition; viruses with maintained or improved functional capacities that have evaded the host immune system survive, while others might go extinct. Thus, the history of virus evolution within an individual can be captured by sequencing viruses sampled over time. Here, we present an individual-based model and corresponding user-friendly R package, wavess, that allows users to simulate within-host virus evolution and evaluate different parameters that affect sequence evolution. We tested and validated wavess by showing that the phylogenetic trees reconstructed from simulated sequences match those generated from empirical HIV-1 data. Taken together, we believe that wavess will be a useful tool for the virus evolution research community.

## Introduction

The diversity acquired by virus sequences during long-term evolution within a host provides useful information for applications such as inferring transmission, developing vaccine candidates, and understanding latent virus reservoir dynamics, ultimately with the goal of curing an infected person. However, in many instances, we are limited in the amount of within-host virus sequence data available to study outcomes of interest, and furthermore, these data may not provide insight into potential underlying mechanisms driving virus evolution. A realistic model of within-host virus evolution calibrated to real sequence data can overcome some of these limitations and enable us to, for instance, investigate sequence evolution over time under different biological scenarios, or perform parameter inference via approximate Bayesian computation.

While many genetic simulation software packages exist [1], few are tailored to within-host evolution of viruses. Those that are applicable to virus evolution tend to be individual-based forward simulation models since important biological features such as recombination and selection can be easily incorporated (e.g., the general simulation framework SLiM [2] and the virus-specific simulation framework SANTA-SIM [3]). It is often not straightforward to customize general simulation frameworks for long-term or chronic within-host virus evolution. On the other hand, more specialized packages that are tailored to within-host virus evolution are not focused on explicitly modeling the immune response to infection, for instance detailed modeling of immune epitopes and their cross-reactivity. Furthermore, it is important to validate any given model against real within-host virus sequences, which has been done for simple summary statistics [3,4], but not for more complex statistics such as those generated from reconstructed phylogenies. Finally, while the goal of these tools is to provide researchers with realistic models, they lack pre- and post-processing capabilities that allow for direct incorporation of empirical data into the analysis. Large amounts of empirical data are available for many viruses, and such data can be used to inform model inputs and validate model outputs. Therefore, there is an existing need for a within-host virus sequence evolution simulation framework that is readily informed by empirical data, easy to use, and straightforward to validate against real-world data.

Here, we present the user-friendly R package and corresponding model, wavess, a within-host adaptive virus evolution sequence simulator. wavess allows users to run easily

customizable forward-in-time individual-based models to study within-host virus evolution in long-term or chronic infections. Previously, we developed a similar, but simpler, model with HIV in mind [5], which included the ability to simulate virus sequences optionally including selection, recombination, and latency. In our new implementation, providing a comprehensive R package, `wavess` includes functionality to take empirical data and generate simulation inputs; therefore, with appropriate adaptations to model inputs and parameters, `wavess` could be applied to other chronic infections such as HCV, or to long-term infections of typically non-chronic viruses such as SARS-CoV-2. We also provide functions for output sequence analysis to support model fitting and downstream analyses. Using HIV-1 as an example, we show that, given empirically informed inputs, the model output sequences recapitulate the trends observed in real data for several distinct phylogenetic summary statistics. Below, we first describe the model and package implementation, then delve into a theoretical sensitivity analysis followed by comparison of model output to empirical data, and finally provide detailed methods for the theoretical and empirical analyses.

## Design and implementation

`wavess`, which is heavily inspired by the model developed by Immonen et al. [5], models the within-host evolution of human viruses as a sequence of synchronized generations beginning with one or more founder sequences. Each generation is one full virus life cycle, from infecting a cell to exiting the cell. Within a generation, infected host cells can change state between being active or latent, viruses in active cells can undergo mutation and recombination, and virus fitness may change based on selective pressures (Fig 1). The next generation of infecting viruses is sampled based on virus fitness. Event counts (e.g., latent cell dynamics,

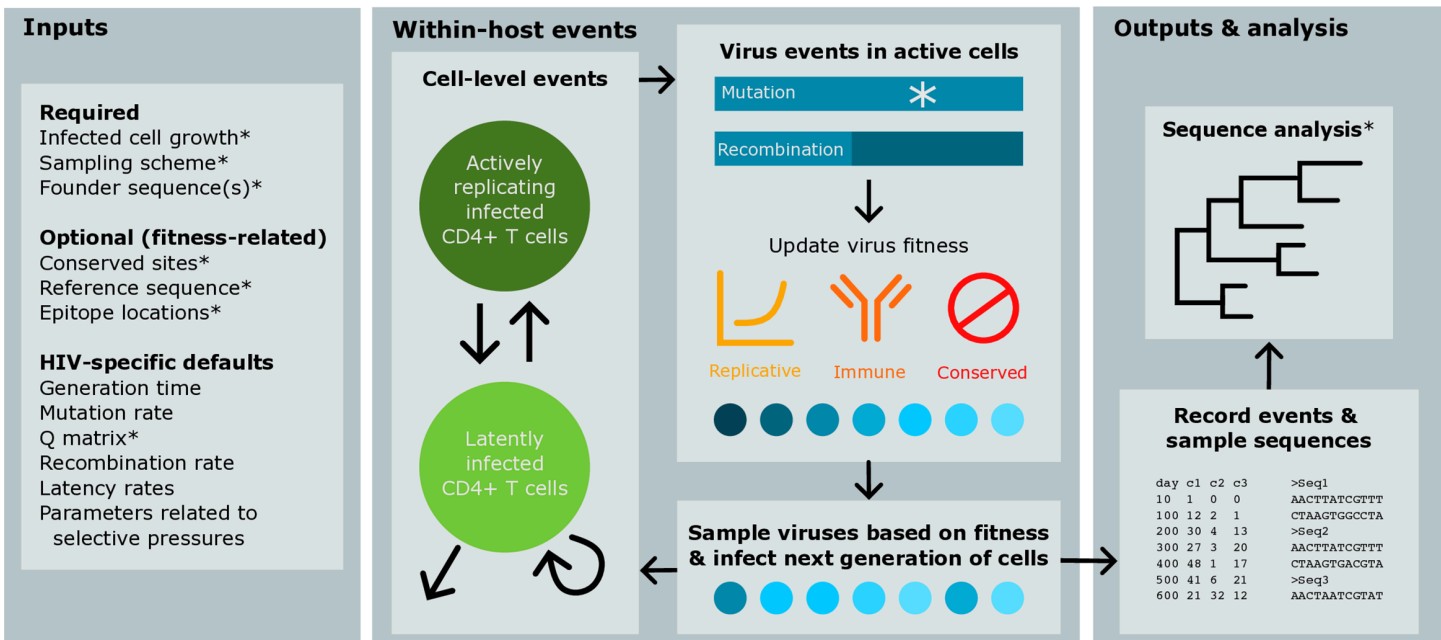

*Function(s) provided to generate input/analyze output

**Fig 1. Overview of the `wavess` algorithm and R package.** The left panel describes required and optional inputs, the middle panel describes the algorithm itself, and the right panel describes the model output and potential analysis.

mutation events, cells with a recombination event) and virus sequences are sampled according to a specified sampling scheme.

## Infected cells

Two populations of infected cells are tracked: an active population and a latent one. The active population can be initialized with one or more infected cells, each containing a virus founder sequence; all founder sequences must be of equal length. Each infected cell in the active population contains one or two virus sequences that can mutate and recombine, and active cells can produce viruses that infect the next generation of active cells. By default, the active cell population follows a discrete logistic growth curve. The viruses that are selected for the next generation are randomly sampled based on their fitness (see below). The number of latently infected cells in each generation is determined by user-specified rates of death, proliferation, activation (transition from latent to active), and deposition (transition from active to latent). These rates are constant and do not change over time. Viruses in latent cells cannot mutate or recombine, and latent cells do not produce viruses.

## Viruses

Each virus has a nucleotide sequence on which the virus fitness is calculated. Nucleotide sequences must all be the same length and are not allowed to have gaps. In each generation, the viruses in active cells can mutate and recombine. Indels are not modeled.

**Mutation.**  The probability of a particular nucleotide mutating to another is specified by a per-site and per-generation mutation rate and a nucleotide substitution rate matrix $Q$ [6]. The $Q$ matrix can correspond to any desired model of DNA evolution.

**Recombination.**  The probability of a recombination event is specified by a per-site and per-generation recombination rate; therefore, there may be more than one cross-over event between two parent viruses in a single active cell. Cells in which at least one template switch occurs are dually infected. All other cells are singly infected.

## Selection

Three selective pressures are considered in wavess: (i) conserved fitness, selection constrained by a set of conserved sites in comparison to a reference sequence; (ii) replicative fitness, selection driven by replicative ability in comparison to a reference sequence; and (iii) immune fitness, selection imposed by a co-evolving immune response at a set of pre-defined epitope locations. Each of the three forms of selection has a fitness cost associated with it. The fitness cost is set in the range [0,1), where 0 indicates no cost, and 1, which indicates no ability to survive, is not allowed as we require a non-zero probability of survival.

The fitness, $F_j$, of virus $j$ is defined by the product of the fitness of each component:

$$F_j = F_j^C \times F_j^R \times F_j^I,\tag{1}$$

where $F_j^C$ is the conserved fitness, $F_j^R$ the replicative fitness, and $F_j^I$ the immune fitness. We describe each of the components of this equation below.

**Fitness by comparison to a reference sequence.**  Conserved fitness and replicative fitness are computed by comparing nucleotides in the virus to an aligned reference sequence, which can be different for each fitness type. Every site in the aligned founder sequence is labeled either conserved or non-conserved. Conserved fitness dictates that a mutation that causes a difference from the reference nucleotide at a conserved site is deleterious to the virus, thus

simulating purifying selection. Replicative fitness dictates that at non-conserved sites, non-reference nucleotides are considered to be slightly less viable than the reference. The equation used to compute $F_j^C$ and $F_j^R$ is

$$F_j^X = (1 - c)^n, \text{ for any } X \in \{C, R\},  \qquad (2)$$

where $n$ is the number of conserved (non-conserved) sites that differ from the reference sequence for $X = C$ ($X = R$), and $c$ is the per-site cost associated with a difference from the reference sequence [5] (Fig 2A). The `wavess` R package uses a default value of $c = 0.99$ when $X = C$ to impose a high cost of mutations in conserved sites and a default value of $c = 0.001$ [7] when $X = R$. For small $c$, Eq 2 is equivalent to $e^{-cn}$, which has previously been used to simulate replicative fitness [8].

A nucleotide position is never considered for both conserved and replicative fitness. When only replicative fitness is modeled, all positions in the simulated sequence that are not an insertion relative to the reference sequence are included in the fitness calculation for $F_j^R$. However, when both forms of fitness are used, if the position is considered to be a conserved site, then it is not included when calculating replicative fitness. When modeling conserved fitness, if the nucleotide in the founder sequence at a user-defined conserved site is not the same as the expected conserved nucleotide, then it is not considered a conserved site under the assumption that the virus founder sequence was viable within its true host.

**Fitness due to a co-evolving immune response.** We model a co-evolving immune response as part of the host environment, which dynamically adapts to recognize frequent virus amino acid epitopes at pre-defined locations of equal length in the sequence. This immune recognition imposes a fitness cost to viruses which contain that epitope, and the strength of recognition optionally increases over time to allow for modeling processes such as antibody affinity maturation or T-cell clonal expansion. Once a virus epitope is recognized, the memory is retained such that an identical sequence coming up, say from the latent reservoir, is immediately recognized. Further, cross-reactivity results in (weaker) recognition of mutated but similar epitopes.

Once an epitope $i$ reaches a user-defined frequency in the population, the fitness cost due to the immune response, $c_i$, increases linearly each generation $t$ until it reaches a maximum

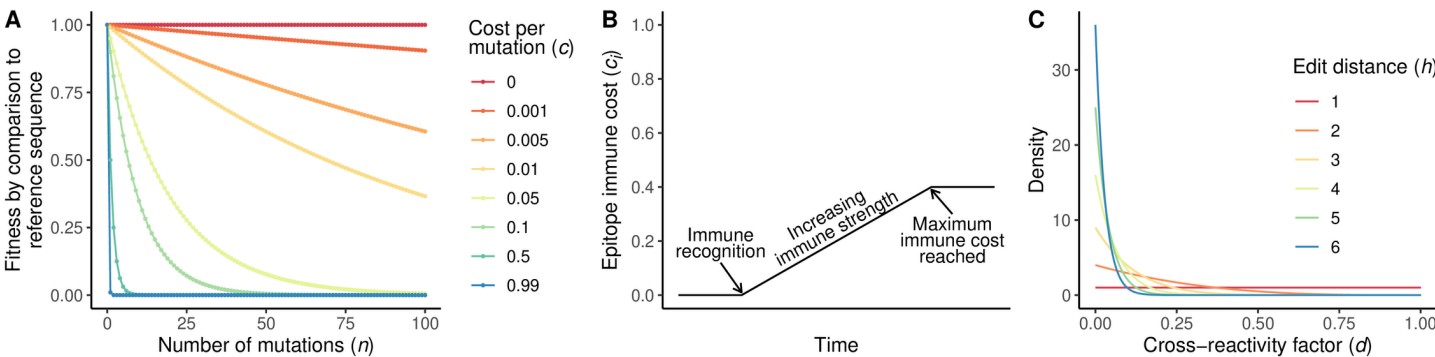

**Fig 2. Visualization of model components related to selection.** (A) Fitness by comparison to a reference sequence (Eq 2) for different values of $n$ (number of mutations) and $c$ (cost per mutation). (B) Depiction of the host immune response to an epitope (Eq 3). Over time, an epitope is recognized once it reaches a certain frequency, followed by a gradual linear increase in the strength of the immune response until a maximum immune cost is reached. (C) Beta distribution from which the reduction in immune cost $d$ is sampled, based on edit distance (number of amino acid mutations) $h$ (Eq 4).

value $c_i^*$ (Fig 2B), as

$$c_i(t) = \min\{c_i^*, c_i^*(t - t_i^0)/g\}, t \geq t_i^0, \tag{3}$$

where $t_i^0$ is the time when the immune response against epitope $i$ is generated and $g$ is the time it takes to reach its maximum.

To model cross-reactivity when a new epitope variant arises in an environment where the host immune system has already mounted an immune response against the virus, we first determine the number of amino acid positions that differ (edit distance) between the new epitope and each of the immune-recognized epitopes. We then identify the cost $c_h$ of the immune-recognized epitope with the minimum edit distance $h$ to the new epitope. The strength of cross-reactivity to the new epitope, $c_{i_x}$, and therefore the immune cost to the new epitope, is the product of $c_h$ and a random value $d$ (the cross-reactivity factor) drawn from a Beta distribution with $\alpha = 1$ and $\beta = h^2$ (Fig 2C):

$$c_{i_x} = c_h\, d. \tag{4}$$

We chose to use a Beta distribution because we assume that larger edit distances between new and recognized epitopes correlate with lower cross-reactivity. The fitness cost of the cross-reactive epitope remains the same across generations unless it reaches the aforementioned user-defined frequency in the population. If this happens, then the immune response evolves according to Eq 3, where the epitope recognition generation $t_i^0$ is based on the cross-reactive immune cost and the original time to maximum immune cost: $t - c_{i_x}g$. This is to maintain the same slope in immune cost increase over time for all epitopes at a given location.

We assume that the immune fitness cost of a virus $j$ is driven by the epitope with the strongest immune response against it, leading to an immune fitness $F_j^I$ of:

$$F_j^I = 1 - \max_{i \in j} c_i, \tag{5}$$

where epitope $i$ must be present in virus $j$.

## Model implementation

The wavess R [9] package includes functions for learning model inputs from empirical sequence data, simulating within-host virus evolution, and generating summary statistics of the output for model evaluation (Fig 1). The back-end of the function that implements the wavess algorithm, run_wavess(), is implemented in Python 3 [10]. We also provide a run_wavess.py file that may be used instead of the R function. The default model parameter values and example inputs included in the package are listed in Table 1 and are tuned to HIV-1 *env* gp120. The R package, Python script, and example data are hosted on GitHub (https://github.com/MolEvolEpid/wavess). The package includes vignettes that describe in detail generating model inputs, running wavess, and analyzing model outputs.

**Inputs.** run_wavess() has three required inputs, each of which can be generated with a helper function:

1. A per-generation infected cell population growth curve (define_growth_curve(); e.g., Fig 3A).
2. A sampling scheme (define_sampling_scheme()).
3. The founder sequence(s) of the infection (extract_seqs()).

**Table 1. Model parameters.**

| Parameter | Default value [range]$^a$ | Unit | Notes/Reference |
|---|---|---|---|
| **Virus** | | | |
| *Sequence features* | | | |
| Founder sequence(s) | HIV-1 *env* gp120 | NA | DEMB11US006 [11] |
| Conserved sites | See Methods | NA | Empirical [12] |
| Replicative reference | HIV-1 subtype B consensus | NA | [12,13] |
| *Population* | | | |
| Generation time | 1 [1 – 2] | Days | [7] |
| Starting population size | 10 [1 – 1000] | Infected cells | |
| Maximum growth rate | 0.3 [0.01 – 1] | $generation^{-1}$ | |
| Maximum population size | 2000 [1,000 – 10,000] | Infected cells | [7] |
| Mutation rate | $4 \times 10^{-5}$ [$10^{-5} – 10^{-4}$] | $site^{-1} generation^{-1}$ | [14] |
| Q matrix | See Methods | Unitless | [15] |
| Recombination rate | $1.5 \times 10^{-5}$ [$10^{-6} – 10^{-4}$] | $site^{-1} generation^{-1}$ | [16] |
| *Individual fitness* | | | |
| Conserved cost | 0.99 [0 – 0.99] | Unitless | |
| Replicative cost | 0.001 [0 – 0.003] | Unitless | [7] |
| Immune cost | 0.3 [0 – 0.9] | Unitless | Fitted |
| **Host** | | | |
| *Immune response* | | | |
| Epitope location | See Methods | NA | Empirical [12] |
| Number of epitopes | 10 | NA | Empirical [12] |
| Epitope amino acid length | 10 [1 – 20] | NA | Empirical [12] |
| Threshold epitope count | 100 [1 – 1000] | NA | |
| Time to maximum immune cost | 90 [1 – 10,000] | Days | [17] |
| *Latency rates* | | | |
| Deposition rate | 0.001 [0.0001 – 0.01] | $day^{-1}$ | |
| Activation rate | 0.01 [0.001 – 0.1] | $day^{-1}$ | |
| Homeostatic proliferation rate | 0.01 [0.001 – 0.1] | $day^{-1}$ | [18] |
| Latent death rate | 0.01 [0.001 – 0.1] | $day^{-1}$ | Set to equal proliferation rate |

$^a$ Default value as of wavess v1.0.0; range is what is investigated in the sensitivity analysis of this report.

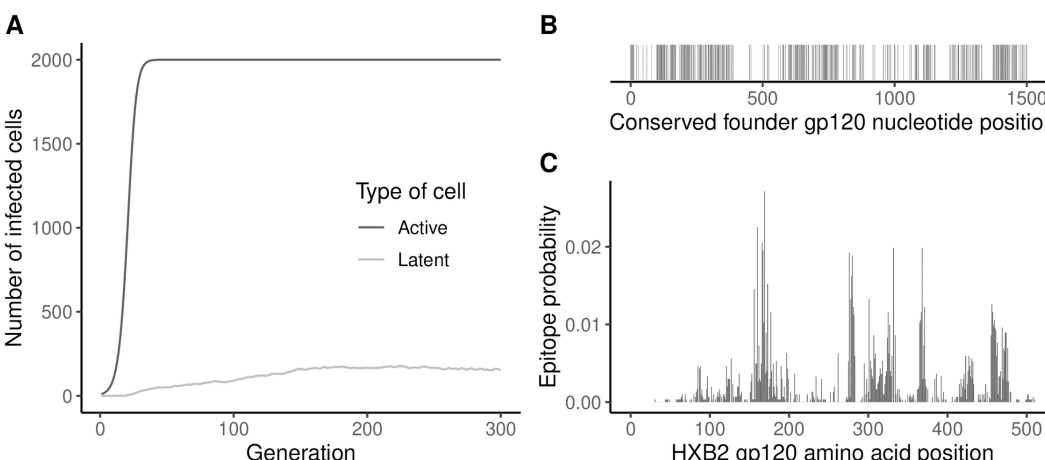

**Fig 3. Visualization of example model inputs and empirical data used to inform model inputs.** (A) Example active (deterministic) and latent (stochastic) growth curves based on default values. (B) Conserved nucleotide sites for the example founder sequence (gp120 of DEMB11US006). (C) Epitope probability distribution for HXB2 gp120. After being converted to nucleotides, these positions can be mapped to the founder sequence.

An example HIV founder sequence (gp120 of DEMB11US006 [11]) is included in the package, but users can also extract a founder sequence from their own alignment or simply provide their own founder sequence.

By default, no selective pressures are modeled. Rather, there are optional inputs corresponding to modeling the three different types of selective pressure: conserved, replicative, and immune. We provide helper functions to generate these inputs based on empirical data:

1. `identify_conserved_sites()` takes as input an alignment of the genomic region to be simulated that is representative of virus diversity, and provides conserved sites and a consensus sequence that can be used as the reference sequence for modeling replicative fitness.
2. `sample_epitopes()` provides epitope locations randomly sampled from an epitope probability distribution that can be generated with `get_epitope_frequencies()`, which takes as input known amino acid epitope locations along the sequence.

The example data provided for modeling virus fitness are specific to the example founder sequence, and include a vector of conserved sites (Fig 3B), the HIV-1 subtype B consensus sequence from the Los Alamos National Laboratory (LANL) HIV database [12,13], and ENV amino acid epitope locations (Fig 3C), also from the LANL HIV database [12].

**Outputs and analysis.** The model outputs event counts, the sampled sequences, and the fitness of each sampled sequence. We also provide the function `calc_tr_stats()` to calculate summary statistics based on a phylogeny reconstructed from the output sequence data. We focus on calculating summary statistics that capture various expected characteristics of the phylogenies of chronic virus infections. Note, however, that other summary statistics might be more useful depending on the question(s) investigated. The phylogenetic summary statistics calculated include:

1. Mean branch length. This measures the phylogenetic inter-node genetic distance, affected by the overall evolutionary rate of the simulated evolutionary system.
2. Mean change in diversity per unit time, measured as the change in within-timepoint tip-to-tip distance across time (slope) calculated using linear regression. This captures how the virus population diversity has changed over time.
3. Mean change in divergence per unit time, measured as the change in root-to-tip distance across time (slope) calculated using linear regression. This captures how virus divergence from the root state has changed over time.
4. Mean leaf depth, measured as the Sackin index normalized by the number of tips in the phylogeny (`treebalance::avgLeafDepI()`) [19]. This captures how ladder-like the tree is, which may be indicative of the strength of selection.
5. Mean number of lineages through time, calculated as the maximum parsimony score (`phangorn::parsimony()`) based on ancestral reconstruction of tree tip labels indicating sampling timepoints [20], normalized by the number of timepoints minus one. This estimate of the mean number of phylogenetic lineages that survived from one sampling to the next is another potential indicator of selection strength.

`calc_tr_stats()` also takes a branch length threshold, which is used to collapse branches less than that value into hard polytomies prior to computing the mean leaf depth. This allows the user to reduce bias in the leaf depth calculation due to zero or very short

branches, which have no information about the bifurcation pattern but still influence the leaf depth calculation.

**Dependencies.** The R package dependencies include the Python packages `numpy` [21] and `scipy` [22], and the R packages `ape` [23], `phangorn` [20], `treebalance` [19], `reticulate` [24], `dplyr` [25], `tibble` [25], and `tidyr` [25]. For plotting in the vignettes, `ggplot2` [25] and `ggtree` [26] are used. For the Python implementation, the `pandas` [27] and `Bio` [28] packages are also required.

## Results

### Summary statistics are sensitive to input parameters

Model sensitivity to input parameter values is dependent on the model output of interest, which is defined by the research question at hand. As an example, we investigate the impact of 14 numeric input parameters on the HIV-1 phylogeny resulting from sequences sampled from the active cell population during ten years of infection. To do so, we first generated 42 points in parameter space using Latin Hypercube Sampling (LHS) with values across the ranges defined in Table 1. As expected, these different points resulted in a wide range in the number of mutation, recombination, and latency events, and in virus fitness (Fig 4A).

We next used the LHS points to generate trajectories through parameter space based on the elementary effects sensitivity analysis method [29] with a perturbation size of 20% of the parameter range. To account for model stochasticity, we computed a normalized mean elementary effect, $\mu^{**}$, where the between-point difference for a parameter is divided by the within-point difference across replicate simulations (see Methods). For some pairs of parameters and summary statistics, there is a similar amount of within-point versus between-point variability (Fig 4B and S1 FigA), indicating that, on average, we do not observe a difference in the summary statistic when that parameter value is perturbed. For other pairs, the between-point difference is larger than the within-point difference (Fig 4B and S1 FigA), indicating that those summary statistics are sensitive to those parameter values. Furthermore, while the correlation between $\mu^{**}$ and the mean between-point difference is strong across summary statistics (S1 FigB), there is a weaker correlation between $\mu^{**}$ and the partial rank correlation coefficient (PRCC) (S1 FigC). This difference may be because $\mu^{**}$ accounts for non-monotonic effects of parameter perturbations on summary statistic values, but PRCC does not.

In these HIV-1 infected host simulations, the parameters with the most influence on the phylogeny were mutation rate and immune cost (Fig 4B). Mean branch length, and mean annual change in diversity and divergence, all branch length statistics, were most sensitive to mutation rate (partial rank correlation coefficient [PRCC] = 0.69, 0.55 and 0.79, respectively), with a 20% change in parameter value across the range tested leading to a greater than two-fold change over stochastic effects. The mean number of lineages through time was most sensitive to immune cost (PRCC = −0.70), with a 20% change leading to a greater than three-fold change in mean number of lineages through time relative to stochastic effects. Other parameters for which a summary statistic was at least two-fold higher with a perturbation change compared to no change include epitope length, days to maximum immune cost, and recombination rate. Changing other parameters resulted in less than a two-fold change in all summary statistics, although mean branch length was generally impacted the most. Average leaf depth, a metric of tree imbalance, was not very sensitive to input parameters under the sampling scheme we used here, although this may be due to large within-point stochasticity as the correlation between average leaf depth and immune cost was quite strong (PRCC = 0.65).

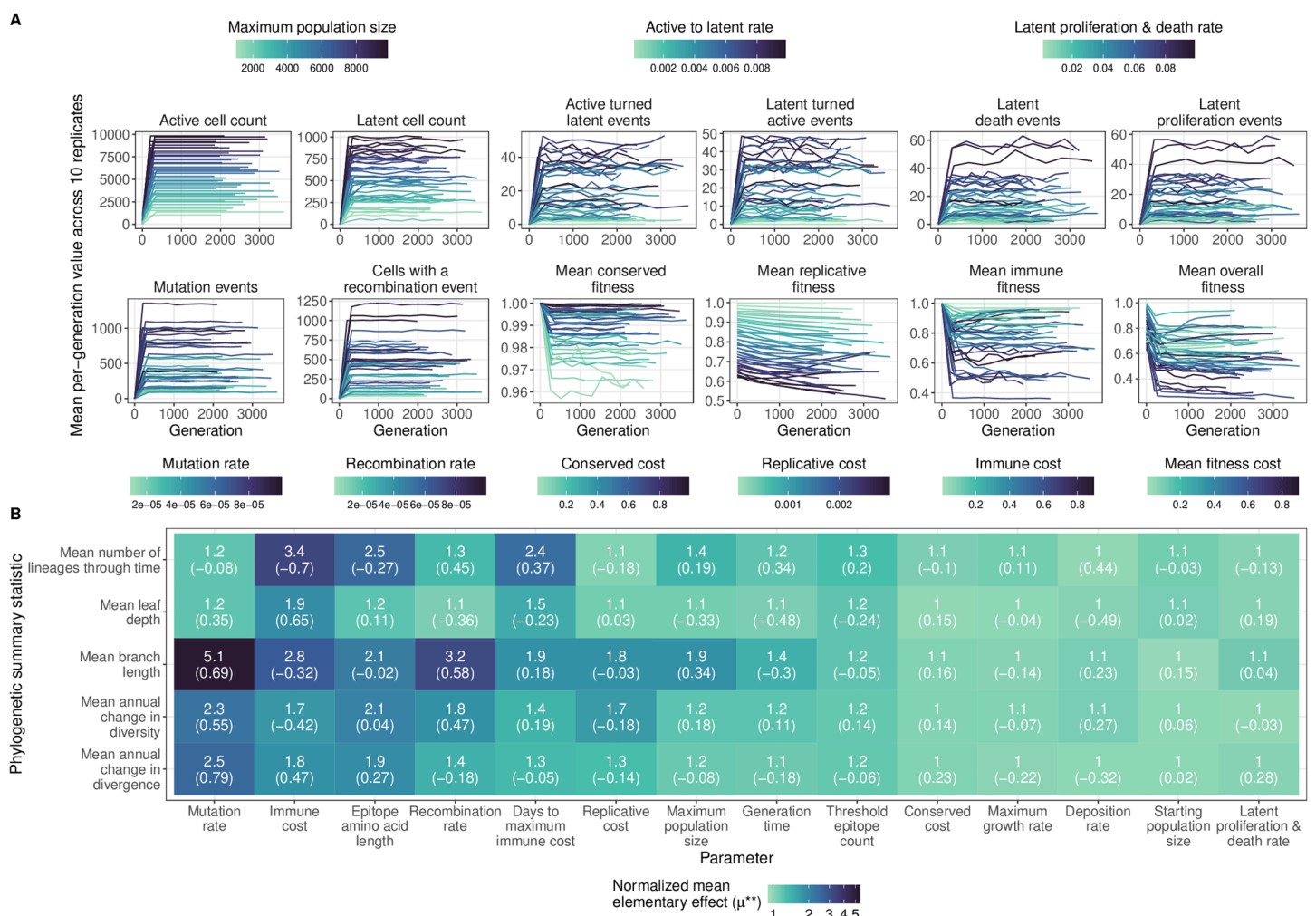

**Fig 4. Sensitivity of model output to input parameters.** (A) Event counts and fitness over time colored by the most relevant parameter value. (B) Normalized mean elementary effect ($\mu^{**}$) of a 20% change in input parameter values across the range in Table 1 relative to stochastic noise for five phylogenetic summary statistics. The partial rank correlation coefficient is shown in parentheses.

## Model output matches empirical data

The usefulness of `wavess` depends on how well it recapitulates empirical data for the research question of interest. As an example, we investigated how well we can match our model output to empirical HIV-1 *env* sequences sampled longitudinally from infected individuals [30,31], and whether this can shed any insights into the immune response of the individuals. We carried out this analysis bearing in mind that virus evolution in these individuals was also a stochastic event with only one realization. For this analysis, we used the functions provided in the `wavess` R package to generate dataset-specific inputs, including founder sequences and sampling schemes. To investigate the host immune response, we varied the immune cost and epitope locations across simulations (see Methods). To evaluate the model, we used the normalized phylogenetic summary statistics described above, and computed the normalized sum of squared errors (SSE) between the phylogeny reconstructed

from the empirical data and the phylogenies of each of the simulated sequence alignments. The summary statistics are not strongly correlated with each other in the empirical data (S2 Fig).

We found that selecting different epitope locations did not significantly affect the phylogenetic summary statistics even when accounting for immune cost (S3 Fig). Furthermore, we observed very little difference between simulations with an immune cost of 0 and 0.1, and between simulations with an immune cost of 0.6 and higher; however, summary statistic values differed in the 0.1-0.6 range (Fig 5A and S4 Fig). Within the 0.1-0.6 range, the empirical summary statistic value fell into the range of simulated summary statistic values for all datasets and summary statistics (n = 55; S4 Fig). For most datasets, we observed one immune cost with a lower SSE than the rest, and the value of the immune cost with the lowest SSE differed across individuals (Fig 5B and S5 Fig). This suggests that the selection pressure imposed by the immune response had different strengths among the individuals studied, which mirrors the previously observed heterogeneous immune response among people [32–34].

We next took a subset of the simulations including only the top 5% based on normalized SSE for each dataset (n=30 per dataset). In this subset, we again observed that the distribution of immune costs varied by individual (Fig 6A). Across all 11 individuals, the empirical summary statistics were on average 0.78 standard deviations away from the mean simulated value (range: [0.01, 2.36]) (Fig 6B). Furthermore, the summary statistic from the best-fitting simulation result and the empirical data were often very similar (Fig 5B). Finally, for the empirical compared to the best-fitting simulated data, the distribution of rate heterogeneity across nucleotide sites estimated using empirical Bayesian methods had similar shapes with long right tails, but the empirical data had a lower median rate (0.52 versus 0.76) and a higher maximum rate (25.2 versus 12.62) (S6 Fig). Empirical phylogenies and rate heterogeneity distributions of representative datasets with an estimated low, medium, and high immune

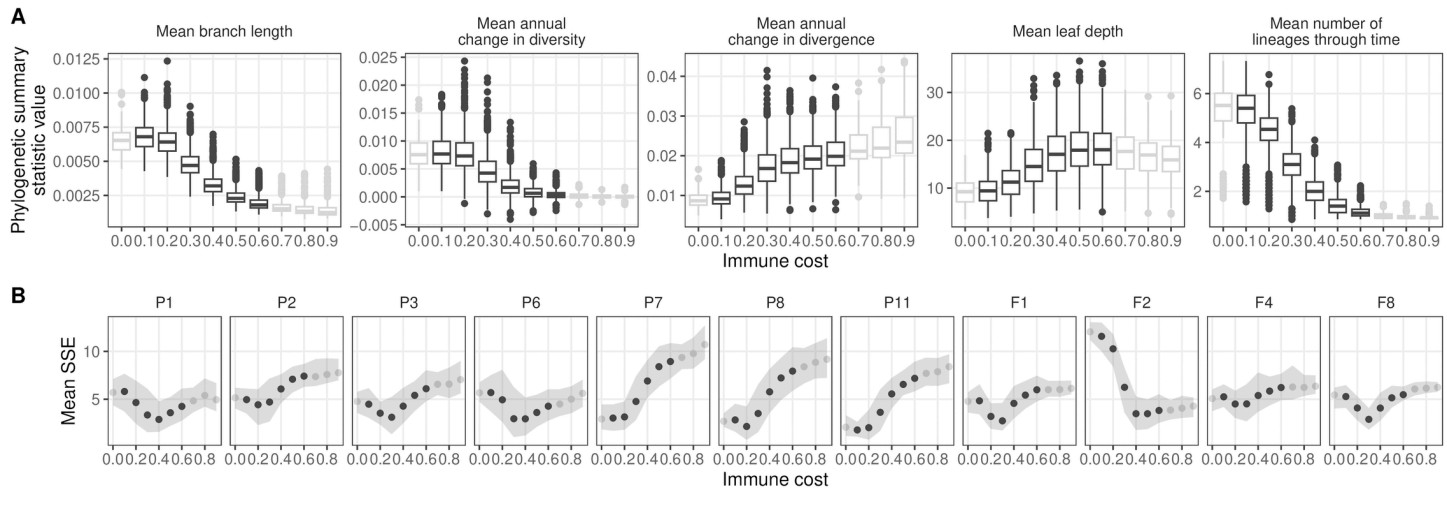

**Fig 5. Summary of simulations performed using empirical founder sequences and sampling schemes with varying immune cost.** (A) Phylogenetic summary statistics for each immune cost. (B) Mean sum of the squared errors (SSE) of the simulated data relative to the empirical data for each immune cost. Color indicates number of replicates performed for a given immune cost.

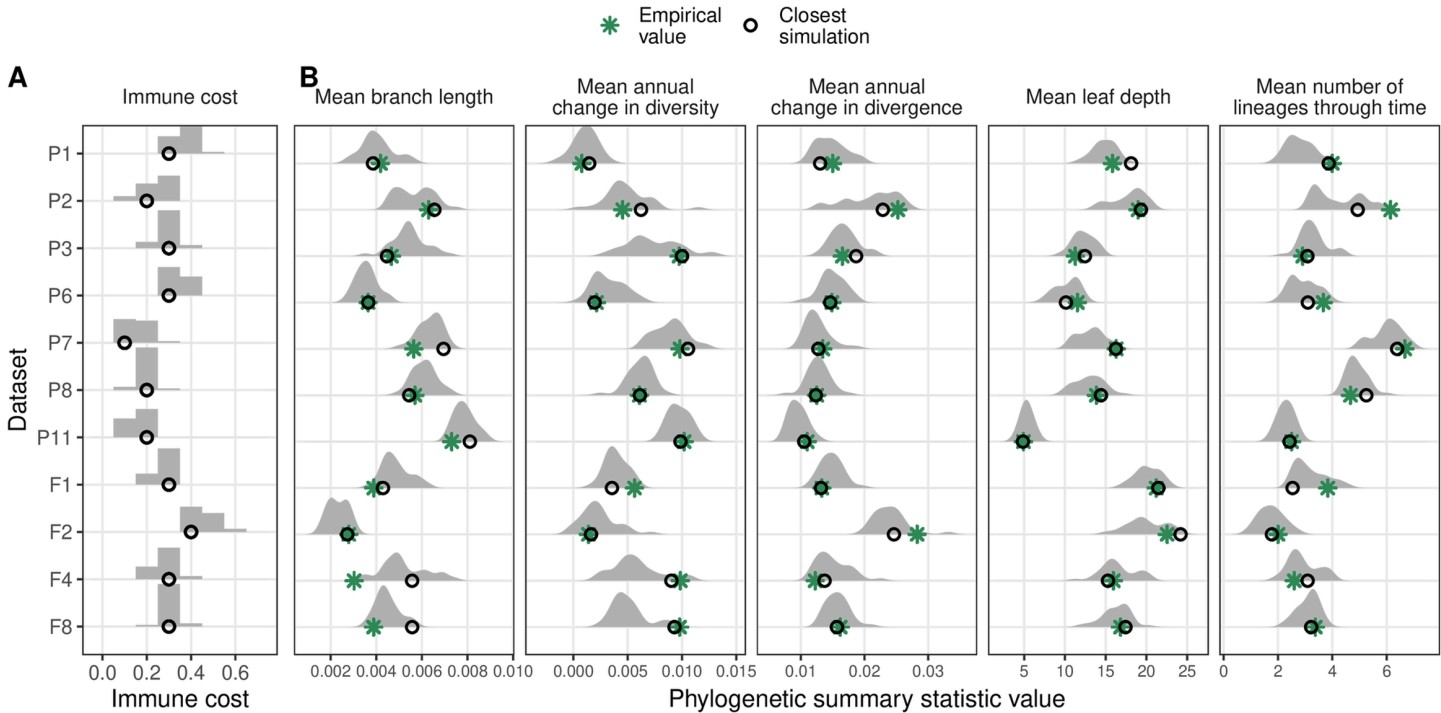

**Fig 6. Summary of top 5% of simulation results based on sum of the squared errors between the empirical and simulated data (*n*=30 per dataset).** (A) Distribution of immune costs. (B) Distribution of phylogenetic summary statistics. Open black circles indicate the best-fitting simulated data and green asterisks indicate the empirical data value.

cost are shown in Fig 7, together with the phylogeny and site rate heterogeneity distribution from the best-fitting simulated data for that sampling scheme.

These findings suggest that the within-host virus evolution simulations generated by wavess can recapitulate empirical data, and can provide insights into the evolutionary parameters that impact HIV-1 within-host evolution.

## Availability and future directions

wavess is hosted on GitHub (https://github.com/MolEvolEpid/wavess). Both the R package and the Python script are included in the repository. While we believe that the current implementation of wavess will allow researchers to answer many biological questions of interest, there is also substantial room to extend the functionality of the package even further. Ways in which wavess could be extended in the future include: (1) allowing for the simulation of segmented viruses, (2) tracking recombination events and virus ancestors and descendants to enable the generation of ancestral recombination graphs and genealogies, (3) dividing the immune response into separate antibody and cytotoxic T cell-like immune responses, (4) adding evolution specific to drug resistance, (5) allowing latent cells to have different lifespans, (6) allowing for compartmentalization of viruses into different tissues, and (7) modeling indels. Furthermore, wavess could be embedded into epidemiological-level models of transmission (e.g. [36]) to make them more realistic. Taken together, we envision that wavess will be an extremely useful tool for the virus evolution research community.

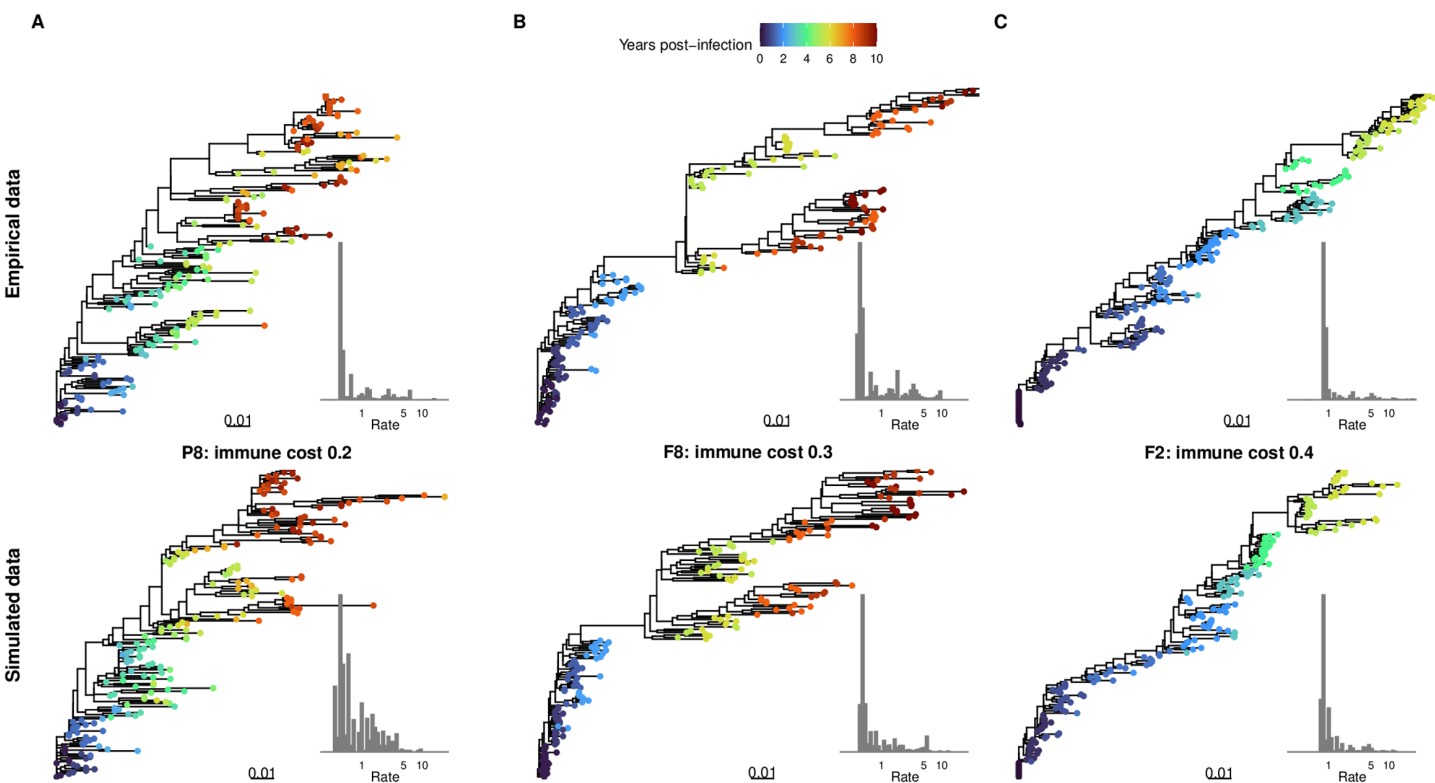

**Fig 7. Phylogenies reconstructed from empirical (top) and simulated (bottom) HIV-1 within-host sequences.** For representative datasets with an estimated (A) low, (B) moderate, or (C) high immune cost. The color indicates the estimated sampling time from infection. The scale bar (0.01) is in units of substitutions/site. The inset for each plot shows the distribution of site-rates estimated by IQ-TREE using an empirical Bayesian method [35].

## Methods

### Elementary effects sensitivity analysis modified for stochastic simulations

We followed the standard elementary effects one-at-a-time global sensitivity analysis algorithm to generate points (sets of 14 parameter values) in parameter space and trajectories of those points [29]. We performed replicate simulations for each point to account for stochasticity in model output. Specifically, we generated 42 trajectories consisting of 15 points each and performed 10 replicate simulations for each point along the trajectory. To ensure broad coverage of the parameter space, the initial 42 points were generated using LHS [37] on the 0-1 range with the R function `lhs::optimumLHS()` [38]. Note that LHS assumes that the input parameter distributions are uncorrelated, which we believe is a reasonable assumption for the parameters used in the sensitivity analysis here. For a given trajectory, the order in which parameters were perturbed was selected at random and each parameter was perturbed by ±0.2. The perturbation direction was selected randomly, and if the resulting value was outside the range 0-1 then the opposite direction was selected. Values from LHS followed by perturbation were linearly transformed from the 0-1 range to the true parameter value using the ranges indicated in Table 1.

For parameter values that were not perturbed, the default values were used (Table 1). For each simulation, 20 samples were taken from the active cell population at generation 5 and each year post-infection, and a maximum-likelihood phylogeny was inferred from

each resulting simulated sequence alignment using IQ-TREE 2 [35] with the GTR+I+R nucleotide substitution model. Phylogenies were rooted on the founder sequence using `phytools::reroot()` [39]; the founder sequence was then dropped from the tree. For each collapsed rooted phylogeny, we computed the five summary statistics described above using `wavess::calc_tr_stats()` with a branch length threshold of 1/1503, where 1503 is the length of the founder sequence; these were the model outputs of interest for the sensitivity analysis.

In a standard elementary effects sensitivity analysis, the revised mean elementary effect $\mu_k^*$ is computed for each parameter $k$, which estimates the impact of the parameter on model output by comparing the output between adjacent points in the trajectory [40]. To extend this to stochastic models, one option is to to calculate $\mu_k^*$ using the mean value of the model output across replicate simulations for each point in parameter space [41]. However, this method does not allow us to determine whether a given parameter change leads to more of a difference than is stochastically observed across replicates at one point in parameter space. Therefore, we instead modified the elementary effects equation by, for each trajectory, normalizing the mean between-point difference in the value of each model output across replicates by the mean within-point difference of the model output across replicates for the original point. Then we calculated the mean normalized value across all trajectories.

The mean between-point difference for the $k^{th}$ of $p$ parameters for a single trajectory $r$ was calculated as:

$$B_{r_k} = \frac{1}{n^2} \sum_{i=1}^{n} \sum_{j=1}^{n} |y(x_1, \dots, x_{k-1}, x_k + \Delta, x_{k+1}, \dots, x_p)_i - y(x_1, \dots, x_{k-1}, x_k, x_{k+1}, \dots, x_p)_j|, \quad (6)$$

where $n$ is the number of replicates (10) and $y$ is a model output. Note that the perturbation size we used here is identical across all simulations (0.2), so we do not need to divide the difference by the perturbation size. This allows for a more straightforward comparison of the normalized values.

The mean within-point difference for parameter $k$ of a single trajectory is given by:

$$W_{r_k} = \frac{2}{n(n-1)} \sum_{i=2}^{n} \sum_{j=1}^{i-1} |y(x_1, \dots, x_{k-1}, x_k, x_{k+1}, \dots, x_p)_i - y(x_1, \dots, x_{k-1}, x_k, x_{k+1}, \dots, x_p)_j|. \quad (7)$$

The normalized mean elementary effect is then calculated as:

$$\mu_k^{**} = \frac{1}{m} \sum_{r=1}^{m} \frac{B_{r_k}}{W_{r_k}}, \quad (8)$$

where $m$ is the number of trajectories (in our analysis, $m=42$). $\mu_k^{**}$ thus allows us to determine whether a given parameter change has more of a difference than is stochastically observed using the original parameter value. Note that the framework described here can be used to investigate sensitivity to any sampling scheme and model output of interest.

We also computed the partial rank correlation coefficient (PRCC) for the 42 initial points in parameter space obtained from LHS using `epiR::epi.prcc()` [42] to investigate the direction of effect between each parameter and summary statistic. Note, however, that this metric only captures monotonic relationships between the parameter and model output.

## Model inputs and analysis for comparison to empirical data

**Sample information.**   We validated wavess using HIV-1 *env* sequences sampled longitudinally from 11 individuals infected with HIV-1 subtype B ($n = 7$ from [30], $n = 4$ from [31]), describing natural HIV-1 evolution. This number excludes individuals from these studies with viruses containing > 1% nucleotide sequence diversity at the first time point ($n = 3$) because we focus on validating the model with datasets where we have sequences from early in infection, and where the individual was likely infected with only one variant. The individuals included in our analysis were either treatment-naive or were eventually put on relatively ineffective treatment targeting non-ENV proteins. In one patient (P2), we observed a decrease in virus load and an increase in CD4$^+$ T cell counts for the last three sampling times (S7 Fig). As treatment appeared to be working at these times, we removed the corresponding samples from the analysis. We included all other samples, even when patients were being treated, as the treatment did not appear to be working well based on viral load and CD4$^+$ T cell count. Furthermore, the evolution of envelope sequences should not be influenced by these drugs. We customized the simulations for each individual by using person-specific founder sequences and sampling schemes that matched the empirical data for each one. To obtain sampling times, we assumed that the infection time was the midpoint between the last negative and first positive test; this is considered sero-conversion time by the original authors. Sequences and metadata were obtained from the LANL HIV database [12]. All dataset IDs used here match the IDs from the original publications.

**Input sequences.**   To generate person-specific founder sequences, we first chose the longest sequence from the first time point for each individual. In the event of a tie, one sequence was chosen randomly. We aligned this sequence to the HXB2 (GenBank accession number K03455) *env* gene as well as the subtype B sequence DEMB11US006 (GenBank accession number KC473833) [11] using the mafft v7.526 ginsi command [43]. We then trimmed off any portion of the sequence outside the *env* gene. Next, we identified, across all sequences, the smallest (6225, *env* start) and largest (7786) HXB2 nucleotide coordinates within *env*. To ensure consistency across simulations, we chose to simulate this entire portion of *env* (1562 nucleotides). For sequences that did not span this entire region, we filled in leading and trailing missing nucleotides with DEMB11US006 to generate the full founder sequence.

**Conserved sites.**   We considered conserved sites to be HXB2 positions that are identical in > 99% of sequences in the *env* filtered alignment of all HIV-1 subtypes from the LANL HIV database [12], identified using wavess::identify_conserved_sites(). These positions were mapped from HXB2 to each individual founder sequence using wavess::map_ref_founder().

**Reference sequence.**   We used the HIV-1 Subtype B consensus sequence from the LANL HIV database [12,13] as the reference sequence for replicative fitness. This was aligned to the founder sequence using the mafft v7.526 ginsi command [43]

**Epitopes.**   To determine the epitope length to use for our simulations, we identified all linear human gp120 epitopes from the LANL HIV database and used the median length (10 amino acids, or 30 nucleotides) for our simulations.

We used the binding, contacts, and neutralization features from the LANL ENV features database [12] to determine the probability of an epitope occurring at each HXB2 position across the length of the founder sequence using wavess::get_epitope_frequencies(). We then used this distribution to randomly sample 10 non-overlapping epitopes of length 30nt across the sequence for each simulation replicate using wavess::sample_epitopes(). The same set of epitope locations was used for each

set of immune cost simulations, and for each dataset. The start positions were mapped from HXB2 to each individual founder sequence via alignment using `wavess::map_ref_founder()`; each epitope was set to be 30nt long beginning from this start position.

**Parameter values.** As HIV sequence evolution is influenced by the strength of an individual's immune responses [32–34], we varied the immune cost and epitope locations across simulations. Each combination of immune cost (range 0-0.9, $n = 10$) and epitope location ($n = 20$ sets for immune costs of 0 and 0.7-0.9, $n = 100$ for immune costs of 0.1-0.6) was simulated once, yielding 680 simulations per sampling scheme. All other parameters were fixed at the values defined in Table 1. `wavess::define_growth_curve()` with default values was used to define the infected cell growth curve.

**Phylogenies.** We first removed leading and trailing segments of the simulated sequences that originated from DEMB11US006, thus keeping only the portion of the simulated sequences derived from the empirical sequence. Then we built phylogenetic trees for each set of empirical and simulated sequences using IQ-TREE 2 [35] with the GTR+I+R model of sequence evolution and the `-rate` option to obtain empirical Bayesian estimates of site rates [44]. Phylogenies were rooted on the sequences from the first timepoint using `phytools::reroot()` [39]. We then calculated the five summary statistics described above for each tree using `wavess::calc_tr_stats()` with a branch length threshold of $1/L_f$, where $L_f$ is the length of the original founder sequence. Next, we computed the difference between phylogenies representing the empirical sequences to the phylogenies reconstructed from each of the sets of simulated sequences by calculating the sum of the squared errors between the empirical and simulated data for centered and scaled values of the five phylogenetic summary statistics.

## Data analysis and visualization

We used the Snakemake (v7.32.4) workflow manager [45] to develop an analysis pipeline and parallelize simulations. All data analysis and visualization was performed in R v4.3.3 [9] with the following packages (in addition to `wavess`): ape v5.8 [23], `phytools` v2.1.1 [39], `tidyverse` v2.0.0 [25], epiR v2.0.78 [42], `ggcorrplot` v0.1.4.1 [46], ggridges v0.5.4 [47], `ggtree` v3.10.0 [26], `treeio` v1.26.0 [48], `scales` v1.3.0 [49], and `ggpubr` v0.6.0 [50].

## Supporting information

**S1 Fig. Comparison of metrics to quantify the sensitivity of a model output (facets) to parameter (color) perturbations.** (A) Mean within-point difference in model output across replicates ($\mu^{**}$ denominator) compared to mean between-point difference with a parameter perturbation ($\mu^{**}$ numerator). Black line is y = x. (B) Normalized mean elementary effect ($\mu^{**}$) compared to mean between-point difference with a parameter perturbation ($\mu^{**}$ numerator). (C) Normalized mean elementary effect ($\mu^{**}$) compared to the partial rank correlation coefficient. R indicates Pearson correlation coefficient, $\rho$ indicates Spearman correlation coefficient.
(TIF)

**S2 Fig. Spearman correlation between phylogenetic summary statistics.** Based on 11 maximum-likelihood phylogenies reconstructed from empirical data.
(TIF)

**S3 Fig. Comparison of phylogenetic summary statistics for different sets of epitope locations.**
(TIF)

**S4 Fig. Phylogenetic summary statistics.** For each real dataset (red line) and simulated dataset across immune costs (black point).
(TIF)

**S5 Fig. Mean sum of squared errors between simulated and empirical data.** For each dataset and phylogenetic summary statistic across immune costs.
(TIF)

**S6 Fig. Distribution of nucleotide site rates for each dataset.** Calculated by IQ-TREE using an empirical Bayesian method where, for each site, the posterior mean site rate across rate categories is weighted by the posterior probability of the site being in each category. The top row is the rates estimated from real sequence alignments and the bottom row is rates estimated from the best-fitting simulated data.
(TIF)

**S7 Fig. Viral load and CD4 count over time for individuals included in the analysis.** The dotted line indicates when an individual was no longer treatment-naive. Samples removed prior to the analysis are indicated as open circles on the plot.
(TIF)

## Acknowledgments

We thank Macauley Locke and Ruian Ke for discussions about the sensitivity analysis.

## Author contributions

**Conceptualization:** Narmada Sambaturu, Zena Lapp, Thomas Leitner.

**Data curation:** Zena Lapp, Fernando D. K. Tria.

**Formal analysis:** Zena Lapp.

**Funding acquisition:** Narmada Sambaturu, Zena Lapp, Thomas Leitner.

**Investigation:** Narmada Sambaturu, Zena Lapp.

**Methodology:** Narmada Sambaturu, Zena Lapp, Fernando D. K. Tria, Ethan Romero-Severson, Carmen Molina-París, Thomas Leitner.

**Software:** Narmada Sambaturu, Zena Lapp.

**Supervision:** Ethan Romero-Severson, Carmen Molina-París, Thomas Leitner.

**Visualization:** Zena Lapp.

**Writing – original draft:** Narmada Sambaturu, Zena Lapp.

**Writing – review & editing:** Narmada Sambaturu, Zena Lapp, Fernando D. K. Tria, Ethan Romero-Severson, Carmen Molina-París, Thomas Leitner.

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
