## [Decision Letter · Decision Letter 0]

29 Jul 2025

PCOMPBIOL-D-25-01084

wavess: An R package for agent-based simulation of within-host virus sequence evolution

PLOS Computational Biology

Dear Dr. Leitner,

Thank you for submitting your manuscript to PLOS Computational Biology. After careful consideration, we feel that it has merit but does not fully meet PLOS Computational Biology's publication criteria as it currently stands. Therefore, we invite you to submit a revised version of the manuscript that addresses the points raised during the review process.

Please submit your revised manuscript within 30 days Sep 28 2025 11:59PM. If you will need more time than this to complete your revisions, please reply to this message or contact the journal office at ploscompbiol@plos.org. Please include the following items when submitting your revised manuscript:

We look forward to receiving your revised manuscript.

Kind regards,

Jordan Douglas

Academic Editor

PLOS Computational Biology

Natalia Komarova

Section Editor

PLOS Computational Biology

**Additional Editor Comments:**

Thank you for your interesting, relevant, and highly-polished mansucript; which has met positive feedback from all three Reviewers. Please work through these very minor revisions proposed by the Reviewers. I look forward to seeing the revised manuscript.

Jordan Douglas

**Journal Requirements:**

3) We notice that your supplementary Figures are included in the manuscript file. Please remove them and upload them with the file type 'Supporting Information'. Please ensure that each Supporting Information file has a legend listed in the manuscript after the references list.

4) Please amend your detailed Financial Disclosure statement. This is published with the article. It must therefore be completed in full sentences and contain the exact wording you wish to be published.

2) If any authors received a salary from any of your funders, please state which authors and which funders..

**Reviewers' comments:**

Reviewer's Responses to Questions

**Comments to the Authors:**

Reviewer #1: This work offers robust computational tools for simulating virus evolution within a host.

Its framework supports recombination, latent infected cell reservoirs, and three types of selection. The system is applied to investigate selection pressures on HIV-1 env sequences, using longitudinal samples from 11 individuals. Validation against real within-host virus data is thorough and rigorous.

Overall, this is a well-constructed and relevant contribution.

comments:

(i) Figure 1, insets (G) and (I), from Grenfell (2004; https://doi.org/10.1126/science.1090727) compare HIV population-level phylogeny with within-host HIV phylogeny. In this study, the authors found that the phylogenies reconstructed from the simulated sequences closely resembled the within-host phylogenies derived from empirical data across all tested summary statistics.

Given these results, would it be feasible to also reconstruct the population-level HIV phylogeny using this simulation framework?

(ii) Latin Hypercube Sampling (LHS) assumes that the input parameter distributions are uncorrelated. Is this assumption realistic in practice?

(iii) second paragraph section 2.3.3: correct "the the".

(iv) section 6.3: The R version v5.3.3 has not yet been released.

Reviewer #2: In the manuscript ”wavess: An R package for agent-based simulation of within-host virus sequence evolution”, the authors describe the titular R package and the model underlying it. In addition, they thoroughly test its sensitivity to varying parameters and compare its results to real data. The biological and mathematical theory behind the model as well as its results are convincing, and the manuscript is very clearly written (and so is the model code behind it). The model itself represents a significant contribution to the growing field of agent-based modelling in pathogen evolution.

I tested the R package using the detailed instructions provided by the authors, and it works as advertised. While I am no expert on R, their code appears on closer reading to do what is stated in the description in the manuscript. The biological assumptions of the model also seem to be well-founded. Finally, the package allows for a large degree of customisation to suit various modelling needs.

My main comments are really more intended as points of discussion or food for thought:

- The authors state that both viruses and their host cells are treated as agents in the model. However, it seems that the infection process itself is not explicitly modelled. Rather, a sample determined through a fitness function is taken from the previous generation of viral sequences and transferred to the following. This is a valid method, but I would not say that it treats the viruses as agents in their own right. As the agency of host cells is also limited, I would almost say that the main agents of the model are the sequences themselves.

- As a consequence of the above, the phylogeny of the model strains cannot be tracked directly, but has to be reconstructed. Again, this is not a problem as such, but one advantage of the agent-based approach can be the ability to directly track relationships between individual agents (e.g., ancestors and descendants). I think a small note on this choice of methodology, possibly in the section on future modelling directions, could benefit the discussion.

Minor comments:

- Drawing a random number from a beta distribution to calculate cross-reactivity seems reasonable, but it would be a good idea with a note on the reasoning behind using this specific distribution.

- The expression given in eq. 3 appears to always give an immune cost of 0 at time t0. However, eq. 4 then describes a starting immune cost. Should this be added to eq. 3?

- It is stated in section 4.2 that 96 % of empirical summary statistics fall within the range of simulated statistics. Since a probability distribution is calculated based on the model results, would it be possible to test whether this observation matches the theoretical probability of outliers?

- In eq. 8, the sum runs over the variable t, but no t appears inside the sum. It would make the equation easier to understand to explicitly show what exactly is summed over.

- On page 10, line 4: “We next subset the simulations to include…”. I found this sentence hard to parse, and suggest replacing it with “We next took a subset of the simulations including…”

- Page 12, line 7: a small grammatical error in “…which parameters was perturbed…”

All in all, I find this model highly interesting and the associated package quite user-friendly. I thank the authors for their contribution to this interesting field.

Reviewer #3: This paper serves as a comprehensive introduction and explanation of a recently implemented R package for simulating within-host evolution of viral sequences. The package simulates models of viral fitness and selection, host immune response, latent reservoirs, and recombination. The model virus that this was built for is HIV but it is applicable well beyond that.

I found the paper easy to follow and well written. The package works as described and has comprehensive vignettes to walk the user through the main functions.

In my view, the paper is publishable as is but could be improved by addressing the following:

0. I think you should make some of the use cases for this software clearer - how would people use it to "allow for the investigation of factors such as the

role of recombination"? (to quote the abstract). An obvious way would be to do parameter inference eg via approximate Bayesian computation or similar - you could mention and reference some works along these lines.

1. in the intro, explain how the paper is structured. Many of the details are relegated to section 6 - it would be good to signal that up front.

2. "Many modeling frameworks" referred to in the intro but only 2 referenced. Add some references here or point to another article where we can find a fuller list.

3. figure 4 needs to be larger to be able to read it, currently lots of detail and a very small font

4. Section 3.2 Page 8: I was confused by "Mean number of transitions per timepoint" as I was unsure what you meant by transition (my mind first turned to transitions vs transversions.) Looking back several pages in the paper I understood but you could remind readers here what you mean.

5. IN the R package, the vignettes are not being found by the system. Eg:

> library(wavess)

> vignette("prepare_input_data")

Warning message:

vignette ‘prepare_input_data’ not found

6. (v minor point) In the vignettes, pipes |> and %>% are both used - choose one (|> works throughout by the looks)

7. With any simulation package, users always ask "but what about...?" In section 5 you discuss a little about future directions but it is not clear how you have structured the implementation to allow others to contribute extensions themselves. THis would be worth a word (or at least thinking about).

**Have the authors made all data and (if applicable) computational code underlying the findings in their manuscript fully available?**

Reviewer #1: Yes

Reviewer #2: Yes

Reviewer #3: Yes

PLOS authors have the option to publish the peer review history of their article (what does this mean?). If published, this will include your full peer review and any attached files.

Reviewer #1: No

Reviewer #2: **Yes: **Andreas Eilersen

Reviewer #3: No

**Figure resubmission:**
---

## [Editor Report · Decision Letter 1]

14 Aug 2025

Dear Leitner,

We are pleased to inform you that your manuscript 'wavess: An R package for simulation of adaptive within-host virus sequence evolution' has been provisionally accepted for publication in PLOS Computational Biology.

Best regards,

Jordan Douglas

Academic Editor

PLOS Computational Biology

Natalia Komarova

Section Editor

PLOS Computational Biology

---

## [Editor Report · Acceptance letter]

PCOMPBIOL-D-25-01084R1

wavess: An R package for simulation of adaptive within-host virus sequence evolution

Dear Dr Leitner,

I am pleased to inform you that your manuscript has been formally accepted for publication in PLOS Computational Biology. Your manuscript is now with our production department and you will be notified of the publication date in due course.

With kind regards,

Zsofia Freund
